# Effect of a Commercial Bentonite Clay (Smectite Clay) on Dairy Cows Fed Aflatoxin-Contaminated Feed

**Antonio Gallo** [1,*] , **Gabriele Rocchetti** [2] , **Fiorenzo Piccioli Cappelli** [1], **Saverio Pavone** [1], **Anna Mulazzi** [1], **Sandra van Kuijk** [3], **Yanming Han** [3] **and Erminio Trevisi** [1]

[1] Department of Animal Science, Food and Nutrition (DIANA), Facoltà di Scienze Agrarie, Alimentari e Ambientali, Università Cattolica del Sacro Cuore, Via Emilia Parmense, 84 29100 Piacenza, Italy; Fiorenzo.piccioli@unicatt.it (F.P.C.); saverio.pavone01@gmail.com (S.P.); anna.mulazzi@unicatt.it (A.M.); erminio.trevisi@unicatt.it (E.T.)

[2] Department for Sustainable Food Processing (DiSTAS), Facoltà di Scienze Agrarie, Alimentari e Ambientali, Università Cattolica del Sacro Cuore, Via Emilia Parmense, 84 29100 Piacenza, Italy; gabriele.rocchetti@unicatt.it

[3] Nutreco R & D, Stationsstraat 77, P.O. Box 299, 3800 AG Amersfoort, The Netherlands; sandra.van.kuijk@trouwnutrition.com (S.v.K.); yanming.han@trouwnutrition.com (Y.H.)

* Correspondence: antonio.gallo@unicatt.it; Tel.: +39-0523-599-433

**Abstract:** We evaluated the impact of dietary supplementation with a commercially available smectite clay (TOXO® MX, Trouw Nutrition, Amersfoort, The Netherlands), that binds to aflatoxins (AFs), on the performance and health status of multiparous lactating Holstein dairy cows that received dietary AFB1 (the main AF). The carry-over of AFB1 was determined by measuring AFM1 (the main metabolite) in dairy milk. Performance values, blood markers, and liver inflammatory markers were also measured. Nine multiparous mid-lactation Holstein cows (parity: 2.67 ± 0.86; days in milk: 91 ± 15 days; milk yield: 40.4 ± 2.7 kg/cow/day) were assigned to one of three treatments in a 3 periods × 3 treatments Latin square design ($n = 3$). In particular, three cows each received the CTR-0 diet (total mixed ration (TMR) with normal corn meals), the CTR-AFLA diet (CTR-0 diet with 17.53 ± 6.55 µg/kg DM AFBI), or the TRT diet (CTR-AFLA diet with 100 ± 1 g/cow/day of smectite clay). The AFB1 level was 0.63 ± 0.50 µg/kg DM in the CTR-0 diet, 2.28 ± 1.42 µg/kg DM in the CTR-AFLA diet, and 2.13 ± 1.11 µg/kg DM in the TRT diet. The experiment consisted of an adaptation period (21 days) and three 17-day experimental periods, each consisting of a 10-day intoxication period and 7-day clearance period. Data were analyzed using the MIXED procedure of SAS (SAS Inst. Inc., Cary, NC, USA) with or without repeated measurements. Overall, the addition of AFB1 reduced the DM intake, but the groups had no significant differences in milk yields. The highest feed efficiency was in the TRT group. Measurement of AFM1 in milk indicated a "plateau" period, from day 4 to day 10 of the intoxication period, when the AFM1 level exceeded the guidelines of the European Union. The commercial smectite clay reduced milk AFM1 concentration by 64.8% and reduced the carry-over by 47.0%. The CTR-0 and TRT groups had similar carry-over levels of AFM1, although the absolute concentrations differed. The groups had no significant differences in plasma biomarkers. These results indicate that the commercially available smectite clay tested here was effective in adsorbing AFs in the gastro-intestinal tracts of cows, thus reducing the excretion of AFM1 into dairy milk.

**Keywords:** mycotoxin binders; mycotoxin binders; Aflatoxin M1; Holstein cows; inflammation; dairy milk quality

## 1. Introduction

Mycotoxin contamination of agricultural products is a serious problem worldwide because the consumption of mycotoxin-contaminated foods can lead to many adverse health effects in humans and animals [1,2]. According to the current limits for mycotoxins established by the European Union (EU) and the *Codex Alimentarius*, the Food and Agricultural Organization (FAO) reported that 25% of worldwide feedstuffs are contaminated by mycotoxins, although there are detectable levels of mycotoxins in 60 to 80% of foods [3]. The major genera of fungi responsible for mycotoxin contamination are *Alternaria*, *Aspergillus*, *Fusarium*, and *Penicillium* [4].

Most mycotoxins are chemically stable compounds that can be detected in animal feed and home-grown forage [5–7]. Ruminants are usually less susceptible than monogastrics to most known mycotoxins because of the combined effects of fiber particles and rumen microflora, which degrade, deactivate, or bind to several mycotoxins [8,9]. Ruminant diets include several starch- and protein-rich fractions with added co-products, and forage, such as silage, haylage, and hay. The conditions used for the preparation and storage of animal feedstuffs, such as grains and silage, favor the growth of fungi responsible for mycotoxin contamination. When a cow eats contaminated feed, its forestomach metabolizes mycotoxins and potentially transfers the product into its milk [10]. Several other factors related to the physiological status of animals or animal husbandry may affect the resistance of ruminants to mycotoxins, such as factors related to peripartum challenges (i.e., reduced immunocompetence, negative energy balance, hypocalcemia, overt inflammation, and oxidative stress) [11]. These factors may impair homeostatic mechanisms, particularly in high-yielding cows, thus increasing the risk of metabolic and infectious diseases and the sensitivity to mycotoxins [12,13].

The aflatoxins (AFs) are a type of mycotoxin produced by *Aspergillus* spp. that are hepatocarcinogenic [14] and contaminants of many foods, including milk and milk-based products [15]. Therefore, interventions are essential to reduce dairy cattle exposure to AFs. The most used mitigation techniques are the selection of seed varieties that have *Aspergillus* spp. resistance, and practicing crop rotations with plants that are not susceptible to *Aspergillus* spp. These methods can reduce the number of infectious spores in the soil [16,17]. The application of chemical agents to animal feed before ensiling can also prevent fungal growth [18,19]. Overall, feed refusal, lethargy, and reproductive disorders are the major symptoms of cattle after acute exposure to AFs. On the other hand, chronic exposure reduces feed efficiency and milk production, and potentially causes and interferes with vaccine-induced immunity [10]. AFs are also responsible for immunosuppression and they have carcinogenic effects on the liver [20,21]. Cattle develop clear signs after the consumption of AFs at concentrations of 1.5–2.2 mg/kg feed, and early indicators may include reduced milk production, photosensitization and, most importantly, reduced immune responses, including reduced responses to vaccination [22]. Because of these subtle effects of AFs, it is difficult to set a safe dietary level; however, the EU legislation states a large difference between the known toxic exposure level (≥1.5 mg/kg feed) and the statutory limit (0.020 mg/kg feed), and this likely provides adequate protection [22].

According to Abrar et al. [23], AFs are toxic because they promote the enzymatic generation of intracellular reactive oxygen species (ROS), such as the superoxide anion, which ultimately lead to the binding of the AF metabolite to DNA, RNA, and proteins. Additionally, AF consumption can lead to increased levels of inflammation-related cytokines and the increased hepatic expression of proteins related to inflammatory responses, including NFKB1 and GPX1 [24–26]. Sequestering agents, such as hydrated sodium calcium aluminosilicate (HSCAS) or bentonite clays, may be added to the diet to reduce the overall impact of AFs on cattle health. HSCAS is a naturally occurring and heat-processed montmorillonite and bentonite is produced from the weathering of volcanic ash and predominantly consists of smectite clay. The smectite group of clays, which includes montmorillonite, is characterized by a 2:1 layered structure and swelling capacity. These agents, or adsorbents, can limit the bioavailability of AFs through ion exchange, and thereby decrease the levels of AF in dairy milk [27,28].

The EU has stricter regulations than the US regarding allowable AF concentrations in milk for human consumption. In particular, the AF level in the EU must be below 0.05 µg/kg in milk,

below 5 µg/kg in complete feedstuffs, and below 20 µg/kg in raw feed [29]. Despite mitigation measures, AF contamination continues to affect the dairy industry and cattle producers. In this regard, the amount of AFM1 (a metabolite of AFB1, the major AF) in dairy milk may represent at least 1–2% of the ingested AFB1; however, several factors can affect this percentage, and high-yielding dairy cows can have AFM1 levels in milk that are even above 6% of ingested AFB1 [22]. For example, model calculations showed that vulnerable high-yield cows given feed that was within the EU limits for AFB1 might produce milk with AFM1 levels above the EU limit. Therefore, people who consume milk or dairy products from high-yielding cows might be exposed to harmful levels of AFM1 [22].

Thus, the objective of this study was to determine the efficacy of a commercially available smectite clay (TOXO® MX, Trouw Nutrition, Amersfoort, The Netherlands) on multiparous lactating Holstein dairy cows exposed to AF challenges on the presence of AFM1 in dairy milk, and on the performance parameters, blood chemistry, and liver inflammatory markers in cows.

## 2. Materials and Methods

The study was authorized by Italian Health regulations that pertain to the accommodation and care of animals used for experimental and other scientific purposes (authorization n. 850/2019-PR issued on 17 December 2019).

### 2.1. Experimental Cows and Diets

Nine multiparous mid-lactation Holstein cows (parity: 2.67 ± 0.86; days in milk (DIM): 91 ± 15; milk yield (MY): 40.4 ± 2.7 kg/cow/day at study onset) were used in experiments conducted at the CERZOO research and experimental center (San Bonico, Piacenza, Italy) starting in January of 2020. The animals were farmed in a tie-stall provided by individual feeding stations and had free access to drinking water. The animals were farmed into three pens, each of which accommodated three cows. Temperature (about 20 °C), relative humidity (about 70%), and duration of daylight (12 h) were maintained in the pens during the experiments. The animals were randomly assigned to three pens before starting the trials.

Before study onset, the animals were allocated to one of three Latin squares based on parity, BW, and MY. The experimental design was a 3 periods × 3 treatments in a Latin square design that consisted of 3 replicates. In each Latin square, three cows received the CTR-0 diet (total mixed ration (TMR) with normal corn meals); the CTR-AFLA diet (TMR with AFB1-contaminated corn meal); or the TRT diet (CTR-AFLA diet with 100 ± 1 g/cow/day of mycotoxin binding product (TOXO MX, Trouw Nutrition, Amersfoort, The Netherlands)).

Each TMR used in this study had the same nutrient composition, and cows were fed once a day at 08:00 h with more than 5% expected orts. The orts were collected individually and weighed daily to determine dry matter intake (DMI) of the previous day. The components (Table 1) were mixed in a mixer wagon (Rotomix 5000, Bravo srl, Cuneo, Italy) in the following order: alfalfa hay, soybean meal (44%), dehulled sunflower meal (34%), salts and mineral–vitamin supplements, corn silage, wheat silage, and water. Then, corn meals (normal diet: AFB1 = 1.76 ± 2.06 µg/kg DM; contaminated diet: AFB1 = 17.53 ± 6.55 µg/kg DM), were added, manually mixed, and fed to animals in dedicated feeding stations. The contaminated corn meal was obtained by in-field crop inoculation with a mycotoxigenic *A. flavus* strain provided by pathologists of Dipartimento di Scienze delle produzioni vegetali sostenibili (Di.Pro.Ve.S.) of Università Cattolica del Sacro Cuore. For cows receiving the TRT diet, 100 g/cow/day of the mycotoxin-binding product was directly added to contaminated corn meals and then mixed into the TMR, according to the procedure recommended by Masoero et al. [30]. The whole TMR with normal or contaminated corn meals was provided individually to each animal based on the measurement of DMI on the previous day, as described above. During the adaptation period and clearance period, all animals received the same TMR diet, which was similar in composition and contained normal corn meals.

**Table 1.** Chemical composition, digestibility, and energy evaluations of experimental total mixed ration (TMR) diets fed to lactating dairy cows.

| Items | Experimental Diets [1] | | | |
|---|---|---|---|---|
| | CTR-0* (*n* = 4) | CTR-0 (*n* = 9) | CTR-AFLA (*n* = 9) | TRT (*n* = 9) |
| **Ingredients (% DM)** | | | | |
| Corn meal | | | 21.7 | |
| Sunflower meal, dehulled 34% | | | 1.9 | |
| Soybean, solvent meal 44% | | | 10.3 | |
| Salts (CaCO$_3$ and NaHCO$_3$) | | | 0.7 | |
| Alfalfa hay | | | 25.8 | |
| Mineral–vitamin supplement [2] | | | 1.1 | |
| Fat (palm oil) | | | 0.8 | |
| Corn silage | | | 31.4 | |
| Wheat silage | | | 6.3 | |
| **Forage:concentrate ratio** | | | 49.9:50.1 | |
| **Chemical composition (% DM)** | | | | |
| DM (% as fed) | 50.6 ± 1.8 | 49.8 ± 2.9 | 50.1 ± 2.4 | 50.8 ± 1.6 |
| Crude protein (CP) | 14.8 ± 0.5 | 14.9 ± 0.5 | 15.1 ± 0.2 | 15.1 ± 0.4 |
| soluble CP | 5.0 ± 0.2 | 4.9 ± 0.3 | 5.0 ± 0.2 | 5.0 ± 0.1 |
| ash | 8.0 ± 0.2 | 8.4 ± 0.5 | 8.5 ± 0.2 | 8.6 ± 0.3 |
| aNDF$_{om}$ | 33.0 ± 1.0 | 32.4 ± 1.5 | 32.4 ± 1.7 | 33.0 ± 1.2 |
| ADF$_{om}$ | 22.2 ± 0.9 | 21.9 ± 1.4 | 22.0 ± 1.5 | 21.7 ± 1.0 |
| ADL | 3.3 ± 0.2 | 3.2 ± 0.3 | 3.3 ± 0.2 | 3.3 ± 0.3 |
| NDFD 24 h | 45.4 ± 1.3 | 46.0 ± 1.3 | 45.1 ± 1.5 | 45.0 ± 0.8 |
| EE | 2.7 ± 0.1 | 2.7 ± 0.2 | 2.9 ± 0.2 | 2.8 ± 0.3 |
| Starch | 23.9 ± 1.8 | 24.3 ± 1.0 | 24.2 ± 1.7 | 24.5 ± 1.4 |
| Sugar | 4.0 ± 0.2 | 4.3 ± 0.3 | 4.1 ± 0.2 | 4.1 ± 0.3 |
| NDICP | 2.1 ± 0.1 | 2.1 ± 0.1 | 2.2 ± 0.2 | 2.3 ± 0.3 |
| ADICP | 0.9 ± 0.1 | 0.8 ± 0.1 | 0.9 ± 0.1 | 0.9 ± 0.1 |
| **Energy evaluations (Mcal/kg DM) [3]** | | | | |
| TDN (%) | 70.1 ± 0.7 | 70.0 ± 0.9 | 70.2 ± 1.1 | 70.1 ± 0.8 |
| ME$_{3x}$ | 2.54 ± 0.05 | 2.54 ± 0.06 | 2.55 ± 0.06 | 2.54 ± 0.05 |
| **AFB1 contamination (µg/kg DM)** | | | | |
| AFB1 in corn meals | 1.76 ± 2.06 | | 17.53 ± 6.55 | |
| AFB1 in TMR | 0.52 ± 0.42 | 0.63 ± 0.50 | 2.28 ± 1.42 | 2.13 ± 1.11 |

[1] CTR-0*: diet including normal corn meal fed to all cows fed during clearance period; CTR-0: diet including normal corn meal; CTR-AFLA: diet including contaminated corn meal; TRT: CTR-AFLA diet with 100 g/cow/day of adsorbent TOXO MX. [2] Mineral–vitamin supplement composition: sodium bicarbonate; 900,000 IU vitamin A; 150,000 IU vitamin D3; 3000 mg vitamin E; 2000 mg encapsulated niacinamide; 20,000 mg niacinamide; 20,000 mg choline chloride; 1100 mg Copper(I) sulfate; 1300 mg MnO; 9400 mg zinc sulfate; 65 mg potassium iodide; 30 mg of sodium selenite; 18,000 mg DL-methionine. [3] Energy evaluations were calculated by using the equations of NRC (2001). Abbreviations: DM, dry matter; aNDF$_{om}$, neutral detergent fiber treated with amylase and sodium sulfite, corrected for residual ash; ADF$_{om}$, acid detergent fiber treated with amylase and sodium sulfite, corrected for residual ash; ADL, acid detergent lignin; NDFD, neutral detergent fiber digestibility; NDCIP, neutral detergent insoluble CP; ADICP, acid detergent insoluble CP; TDN, total digestible nutrient; ME, metabolizable energy.

The whole study lasted 72 days, and consisted of a group formation and adaptation period (21 days), and three 17-day experimental periods, each of which consisted of 10 days of intoxication (in which each cow was assigned to a specific treatment) and 7 days of clearance (in which all animals received the same non-contaminated TMR, as described above).

## 2.2. Analysis of Feeds, Diets and Mycotoxins

Samples of feeds and TMR were taken on day 1, day 5, and day 10 of each intoxication period and on day 3 of each clearance period. Then, the samples were subjected to chemical analysis (i.e., TMR) and measurement of mycotoxin contamination (i.e., TMR and normal or contaminated corn meals), as previously described [31]. Briefly, samples were dried at 60 °C in a ventilated oven for 48 h, milled through a 1-mm screen using a laboratory mill (Thomas-Wiley, Arthur H. Thomas Co., Philadelphia, PA, USA), and then stored for subsequent analysis. Uncorrected DM was determined by the gravimetric loss of free water after heating at 105 °C for 3 h (AOAC method 945.15) [32]. Then, the DM concentration was corrected for volatile losses that occurred during oven drying using equations from NorFor [33]. Ash concentration was determined as the gravimetric residue after incineration at 550 °C for 2 h [32] (AOAC method 942.05), and an ether extract was used to measure crude fats (AOAC method 920.29) [32], and crude protein (CP; N × 6.25) was determined using the Kjeldahl method [32] (AOAC method 984.13). The soluble fraction of CP (expressed on a DM basis) was determined according to Licitra et al. [34]. Neutral detergent (ND), acid detergent (AD), and lignin sulfuric acid (ADL) fiber fractions were sequentially measured using the AnkomII Fiber Analyzer (Ankom Technology Corporation, Fairport, NY, USA), as described by Van Soest et al. [35]. The ND solution contained sodium sulfite and a heat stable amylase (activity: 17,400 Liquefon-U/mL, Ankom Technology). All fiber fractions were corrected for residual ash (aNDFom, ADFom). The starch content was determined by polarimetry (Polax 2 L, Atago®, Tokyo, Japan).

The AFs in corn meals and TMR were measured as described by Gallo et al. [36]. Briefly, AFs were extracted from 10 g of dried feed using 100 mL of an acetone:water solution (70:30 *v/v*), the mixture was shaken at 150 r.p.m. for 45 min (Universal Table Shaker 709), and was then passed through a Schleicher & Schuell 595 1/2 filter paper (Dassel, Germany). Then, a 5 mL aliquot was purified on an immunoaffinity column (Aflatoxin Easi-extract, R-Biopharm Diagnostics Technologies, Glasgow, UK). The column was washed with 5 mL of water and slowly eluted with 2.5 mL of methanol. The eluate was concentrated under a flow of nitrogen and brought to a volume of 2 mL with a solution of acetonitrile:water (41:59, *v/v*), and then filtered into HPLC vials (Millipore Corporation, Bedford, MA, USA; HV 0.45 mm) for subsequent chromatographic analysis. The AFs were separated using a reverse-phase RP-18 Supersher column (5 μm particle size, 125 × 4 mm i.d.; Merck, Darmstadt, Germany) using an isocratic elution of 59:41 (*v/v*) of mobile phase A (water) and mobile phase B (acetonitrile: methanol: 17:29, *v/v*) at a flow rate of 1 mL/min. The AFs were subjected to post-column photochemical derivatization using a UV lamp at 254 nm (UVETM derivatizer, LC Tech, Dorfen, Germany). Then, fluorescence was measured at 440 nm following excitation at 365 nm. Standard stock solutions were used to determine AF concentrations in samples (AOAC 970.44) [32] and were stored at −20 °C when not in use.

## 2.3. Body Weight and Body Condition Score

The cows were weighed at the end of each intoxication period. The body condition score (BCS) was determined before and at the end of each 10 day intoxication period using a 4-point scoring system [37].

## 2.4. Health Status of Cows

All cows were healthy at study onset and health status was monitored daily by the farm's veterinarian throughout the study. Mastitis was diagnosed by visual evaluation of abnormal milk from each quarter, and somatic cell count (SCC) analysis was performed for suspicious cases. Diarrhea was diagnosed by visual evaluation of the consistency and color of feces using the fecal score method [38]; diarrheic feces were those with a fecal score of 2 or less.

Some health problems were recorded during the experiment (2 cases of mastitis and 1 nipple injury), but these events were limited and short-lived. All the cows recovered quickly, thus indicating

good health status. Because of these minor health problems, some data were excluded from statistical analysis (4.1% from the intoxication periods and 2.6% from the clearance periods).

## 2.5. Milk Yield, Composition, and AFM1 Analysis

The milk yield of each cow was recorded at each milking, and representative 100 mL samples were taken at each milking time during the experimental periods. Fat, protein, casein, lactose, titratable acidity, and coagulation properties (rennet clotting time (r) and curd firmness (A30)) were measured using infrared measurements (MilkoScan FT 120, Foss Electric, Hillerød, Denmark) according to Chessa et al. [39]. The daily production of fat, protein, casein, and lactose were then calculated. Milk urea was determined in skimmed milk using a spectrophotometric assay and a urea nitrogen kit (cat# 0018255440, Instrumentation Laboratory, Milano, Italy) with an auto-analyzer (ILAB-650, Instrumentation Laboratory, Lexington, MA, USA). SCC was determined using an optical fluorimetric method with an automated cell counter (Fossomatic 180, Foss Electric) on day 1, day 5, and day 10 of each experimental period.

The milk was analyzed for AFM1 on multiple days (i.e., days 1, 2, 4, 5, 8, 10, 12, 14 and 17) during each experimental period. To quantify AFM1 in milk, extraction was performed using an immunoaffinity technique, according to Mortimer et al. [40]. Briefly, 50 mL of milk were centrifuged at 7000 rpm for 10 min at 4 °C and then passed through Schleicher & Schuell 595 1/2 filter paper (Dassel, Germany). Then, an aliquot of 20 mL was passed through an immunoaffinity column (Aflatoxin Easy-extract, R-Biopharm Diagnostics Technologies, Glasgow, UK). The column was washed with 5 mL of water, and then slowly eluted with 2.5 mL of methanol. The extract was dried under a flow of nitrogen, redissolved in 1 mL of acetonitrile:water (25:75, *v/v*) and then passed through a filter (Millipore Corporation, Bedford, MA, USA; HV 0.45 μm). The purified extract was injected into a Jasco HPLC system with a FP-1520 fluorescence detector (Jasco, Tokyo, Japan). Jasco Borwin Chromatography software was used for system control and data collection. Finally, AFM1 was separated with a reverse-phase RP-18 LiChrospher column (Merck, Darmstadt, Germany; 5 μm particle size, 125 × 4 mm i.d.) at room temperature, using an isocratic elution with water and acetonitrile (75:25 *v/v*) at a flow rate of 1 mL/min. Then fluorescence was measured at 440 nm following excitation at 365 nm. Standard stock solutions of AFM1 were used to determine AFM1 concentrations in samples (AOAC 970.44) [32] and were stored at −20 °C when not in use.

## 2.6. Blood Sampling and Blood Biochemistry

Blood samples were collected for chemical analysis before the start of the trial and on day 11 of each experimental period. The samples were taken in the morning (before feeding) by venipuncture of the jugular vein using 10-mL Li-heparin treated tubes (Vacuette, containing 18 IU of Li-heparin/mL, Kremsmünster, Austria) and immediately cooled in an ice water bath. A small amount of blood (0.07 mL) was used to calculate packed cell volume (PCV; Centrifugette 4203; ALC International Srl, Cologno Monzese, Italy) and the remaining blood was centrifuged (3500× *g*, 16 min, 4 °C), and the resulting plasma (5–6 mL) was separated into aliquots and stored at −20 °C until analysis.

Plasma metabolites were analyzed at 37 °C using an automated analyzer (ILAB 650, Instrumentation Laboratory, Lexington, MA, USA) as described by Calamari et al. [41]. Commercial kits from Instrumentation Laboratory SpA (Werfen, Italy) were used to measure glucose, total cholesterol, urea, Ca, P, Mg, total protein, albumin, total bilirubin, and creatinine. Kits from Wako Chemicals GmbH (Neuss, Germany) were used to measure non-esterified fatty acids (NEFA), beta-hydroxybutyric acid (BHBA), and Zn. Electrolytes (Na, K, and Cl) were measured using a potentiometer (ion-selective electrode connected to ILAB 650). Kinetic analysis was used to determine the activities of alkaline phosphatase (AP; EC 3.1.3.1), aspartate aminotransferase (AST; EC 2.6.1.1), and γ-glutamyltransferase (GGT; EC 2.3.2.2), with kits from Instrumentation Laboratory SpA. Ceruloplasmin and haptoglobin were measured as described by Calimari et al. [41]; paraoxonase (PON) activity as described by Bionaz et al. [42]; myeloperoxidase (MPO) activity as described by Bradley et al. [43]; reactive oxygen

metabolites (ROMt) as described by Jacometo et al. [44]; ferric reducing antioxidant power (FRAP) as described by Benzi et al. [45].

## 2.7. Carry-Over Calculation

The carry-over of AFM1 in milk was calculated daily as described by Masoero et al. [46]. This number indicated the percentage of AFB1 consumed by each animal (µg/cow/day) that was excreted as AFM1 in milk (µg/cow/day).

## 2.8. Statistical Analysis

Variables with non-normal distributions (such as SCC) were log-transformed before statistical analysis. There were three replicates of 3 periods × 3 treatments in a Latin square design.

Data that were measured once in each cow during each experimental period were tested for normality and analyzed using the MIXED procedure in SAS (SAS Inst. Inc., Cary, NC, USA, release 9.3, 2002–2010) according to the model:

$$Y_{ijklm} = \mu + T_i + L_j + p_k + c_l + e_{ijklm}$$

where $Y_{ijklm}$ is the dependent variable, $\mu$ is the population mean, $T_i$ is the fixed effect of treatments, $L_j$ is the fixed effect of the Latin square, $p_k$ is the fixed effect of period, $c_{l(k)}$ is the random effect of a cow, and $e_{ijklm}$ is the residual error.

Data that were measured more than once in each cow during each experimental period were tested for normality and analyzed as repeated measurements using the MIXED procedure in SAS (SAS Inst. Inc., Cary, NC, USA, release 9.3, 2002–2010), according to the model:

$$Y_{ijklmn} = \mu + T_i + D_j + L_k + T \times D_{ij} + p_m + c_n + e_{ijklmn}$$

where $Y_{ijklmn}$ is the dependent variable, $\mu$ is the population mean, $T_i$ is the fixed effect of treatment, $D_j$ is the fixed effect of time of measurement (repeated measurements), $L_k$ is the fixed effect of the Latin square, $TD_{ij}$ is the fixed effect of treatment x time of measurement interaction, $p_m$ is the fixed effect of period, $c_n$ is the random effect of a cow, and $e_{ijklmn}$ is the residual error.

One of the five covariate model structures was used based on the finite-sample corrected Akaike information criterion (AICC) and the Schwarz Bayesian criterion for the best fitting model. The five tested structures were: compound symmetry, heterogeneous compound symmetry, unstructured, autoregressive (1), and ante-dependence [47,48]. A *p*-value below 0.05 was considered significant, and a *p*-value between 0.05 and 0.10 was considered to indicate a trend.

## 3. Results

### 3.1. Chemical Composition, Digestibility, and Energy Evaluation of TMR

Table 1 shows the ingredients and chemical composition of the TMR and the AFB1 levels in the TMR and corn meals. The AFB1 concentration was 1.76 ± 2.06 µg/kg DM in normal cornmeal and was 17.53 ± 6.55 µg/kg DM in contaminated cornmeal. The AFB1 level was 0.63 ± 0.50 µg/kg DM in the CTR-0 diet, 2.28 ± 1.42 µg/kg DM in the CTR-AFLA diet, and 2.13 ± 1.11 µg/kg DM in the TRT diet. No other AFs were detectable.

### 3.2. Feeding Behavior, Milk Yields, Feed Efficiency, and Milk Parameters

Table 2 shows the relationships of diet with feeding behavior, milk yields, feed efficiency, and dairy milk parameters. The DMI was greater (*p* < 0.05) in the CTR-0 group (19.5 kg DM/cow/day) than in the CTR-AFLA group (18.7 kg DM/cow/day) and the TRT group (18.4 kg DM/cow/day). Furthermore, the DMI was lower (*p* < 0.05) during period 3 than during the other periods. The DMI during the clearance period was 2.73% BW and was greater (*p* < 0.05) in the CTR-0 group than in the CTR-AFLA

group (2.62% BW) and the TRT group (2.62% BW) during the intoxication periods. Similarly, the DMI was higher ($p < 0.05$) during period 1 and period 2 than period 3. The MY did not differ among treatments, and the average was 29.0 kg/cow/day. Overall, the MY was higher ($p < 0.05$) during period 1 (30.6 kg/cow/day) than during period 2 (29.1 kg/cow/day) and period 3 (27.1 kg/cow/day). Similarly, the 3.5% corrected FCM and ECM did not differ among treatments, and the averages of the three periods were 30.4 kg/cow/day (FCM) and 31.2 kg/cow/days (ECM). The feed efficiency (Milk/DMI) was different among the three treatments ($p < 0.05$; TRT: 1.61, CTR-AFLA: 1.57, CTR-0: 1.52), and these trends were similar for other feed efficiency measurements (3.5%FCM/DMI and ECM/DMI). Analysis of dairy milk characteristics and quality parameters indicated no differences in the fat, protein, and casein levels among the different groups in terms of milk composition and daily milk excretion. Lactose production tended to be higher ($p < 0.083$) in the CTR-0 group than in the TRT group (1510 vs. 1477 g/cow/day), although this difference was only 2.2%. Coagulation properties, milk urea, and log(SCC) did not differ among the groups. As expected, there were significant differences in key milk quality parameters—protein, fat, casein, lactose, and urea—among the different experimental periods.

**Table 2.** Least squares means and associated standard errors of the mean (pooled SEM) for feeding behavior, milk yields, feed efficiency, and milk parameters of Holstein cows that received CTR-0, CTR-AFLA, and TRT diets during each intoxication period. Values with different superscript letters in the same group and row were significantly different ($p < 0.05$).

| Items | Units of Measurements | Treatment | | | Period | | | Pooled SEM | $p$ of the Model | | | | |
|---|---|---|---|---|---|---|---|---|---|---|---|---|---|
| | | CTR-0 | CTR-AFLA | TRT | 1 | 2 | 3 | | Latin Square | Period | Treatment (T) | Day (D) | D * T |
| **Feeding Behavior** | | | | | | | | | | | | | |
| DMI | kg/cow/day | 19.5 [a] | 18.7 [b] | 18.4 [b] | 18.4 [a] | 19.1 [a] | 17.9 [b] | 0.27 | 0.297 | <0.05 | <0.05 | 0.074 | 0.841 |
| DMI | % BW | 2.73 [a] | 2.62 [b] | 2.62 [b] | 2.75 [a] | 2.69 [a] | 2.53 [b] | 0.005 | 0.109 | <0.05 | <0.05 | 0.070 | 0.857 |
| **Milk yields** | | | | | | | | | | | | | |
| MY | kg/cow/day | 29.2 | 28.8 | 28.8 | 30.6 [a] | 29.1 [b] | 27.1 [c] | 0.63 | 0.530 | <0.05 | 0.472 | <0.05 | 0.963 |
| 3.5% FCM | kg/cow/day | 30.7 | 30.2 | 30.2 | 31.2 | 31.4 | 28.6 | 1.08 | 0.260 | <0.05 | 0.541 | <0.05 | 0.994 |
| ECM | kg/cow/day | 31.6 | 31.0 | 31.0 | 32.2 [a] | 32.1 [a] | 29.4 [b] | 1.01 | 0.999 | <0.05 | 0.475 | 0.072 | 0.997 |
| Milk/DMI | dmnl | 1.52 [c] | 1.57 [b] | 1.61 [a] | 1.57 | 1.55 | 1.57 | 0.001 | 0.524 | 0.454 | <0.05 | <0.05 | 0.996 |
| 3.5%FCM/DMI | dmnl | 1.59 [b] | 1.65 [a,b] | 1.69 [a] | 1.61 | 1.67 | 1.65 | 0.004 | 0.355 | 0.147 | <0.05 | <0.05 | 0.991 |
| ECM/DMI | dmnl | 1.63 [c] | 1.69 [b] | 1.73 [a] | 1.65 [c] | 1.70 [a] | 1.70 [b] | 0.004 | 0.361 | 0.207 | <0.05 | <0.05 | 0.996 |
| **Milk parameters** | | | | | | | | | | | | | |
| Fat | % | 4.27 | 4.30 | 4.31 | 4.09 [b] | 4.48 [a] | 4.31 [a] | 0.036 | 0.856 | <0.05 | 0.888 | <0.05 | 0.615 |
| Fat | g/cow/day | 1243 | 1224 | 1226 | 1238 [b] | 1297 [a] | 1159 [c] | 3538.7 | 0.161 | <0.05 | 0.759 | <0.05 | 0.976 |
| Protein | % | 3.13 | 3.12 | 3.13 | 3.10 [b] | 3.12 [a,b] | 3.15 [a] | 0.002 | 0.106 | 0.065 | 0.822 | 0.380 | 0.953 |
| Protein | g/cow/day | 917 | 897 | 901 | 948 [a] | 910 [b] | 859 [c] | 789.6 | 0.112 | <0.05 | 0.412 | 0.065 | 0.998 |
| Casein | % | 2.38 | 2.38 | 2.39 | 2.36 [b] | 2.38 [b] | 2.41 [a] | 0.001 | 0.129 | <0.05 | 0.820 | 0.251 | 0.878 |
| Casein | g/cow/day | 699 | 686 | 688 | 723 [a] | 694 [b] | 656 [c] | 440.3 | 0.116 | <0.05 | 0.459 | <0.05 | 0.996 |
| Lactose | % | 5.14 | 5.15 | 5.14 | 5.16 [a] | 5.12 [b] | 5.15 [a] | 0.001 | 0.292 | <0.05 | 0.368 | 0.284 | 0.965 |
| Lactose | g/cow/day | 1510 | 1483 | 1477 | 1578 [a] | 1489 [b] | 1404 [c] | 1317.9 | 0.463 | <0.05 | 0.083 | <0.05 | 0.969 |
| Titratable acidity | °SH/50 mL | 3.03 | 3.06 | 3.01 | 3.01 | 3.06 | 3.02 | 0.006 | 0.792 | 0.282 | 0.338 | 0.063 | 0.794 |
| Clotting time, r | min | 18.71 | 18.52 | 18.01 | 18.52 [a,b] | 20.72 [a] | 16.01 [b] | 8.122 | 0.958 | <0.05 | 0.873 | 0.619 | 0.742 |
| Curd firmness, a30 | mm | 29.16 | 29.63 | 29.75 | 29.52 | 29.19 | 29.82 | 0.955 | 0.461 | 0.482 | 0.506 | 0.182 | 0.696 |
| Milk urea | mg/100 mL | 36.43 | 37.14 | 36.94 | 38.11 [a] | 37.11 [a] | 35.29 [b] | 1.652 | <0.05 | <0.05 | 0.618 | <0.05 | 0.577 |
| LogSCC | Log10 (cells/mL) | 2.01 | 2.06 | 1.95 | 2.01 | 2.01 | 2.00 | 0.018 | 0.816 | 0.978 | 0.508 | 0.272 | 0.639 |

Abbreviations: dmnl, dimensionless; DMI, dry matter intake; MY, milk yield; FCM, fat corrected milk; ECM, energy corrected milk; SCC, somatic cell count.

### 3.3. AFB1 Intake and Carry-Over of AFB1 into Dairy Milk as AFM1

The Table 3 shows the relationships of AFB1 intake with milk AFM1 and the carry-over of AFB1 into dairy milk as AFM1. As expected, the AFB1 intake was higher in the CTR-AFLA and TRT groups than in the CTR-0 group (42.461 and 39.197 vs. 11.204 μg/cow/day). During the whole experimental period, the highest ($p < 0.01$) AFM1 concentration was in the CTR-AFLA group and the lowest was in the CTR-0 group (43.16 vs. 8.81 ng/kg milk, $p < 0.01$), and there was an intermediate level in the TRT group (24.01 ng/kg milk). During the "plateau" period (day 4 to day 10 of the intoxication period; Figure 1), the highest ($p < 0.01$) AFM1 concentrations were in the CTR-AFLA group (96.51 ng/kg milk), the lowest concentrations ($p < 0.01$) were in the CTR-0 group (8.90 ng/kg milk), and there were intermediate concentrations in the TRT group (33.96 ng/kg milk). Thus, during the plateau period, TOXO MX supplementation reduced the AFM1 concentration in dairy milk by 64.8%. In addition, the carry-over was about two-fold higher in the CTR-AFLA group than in the TRT group (4.73 vs. 2.51%, $p < 0.05$).

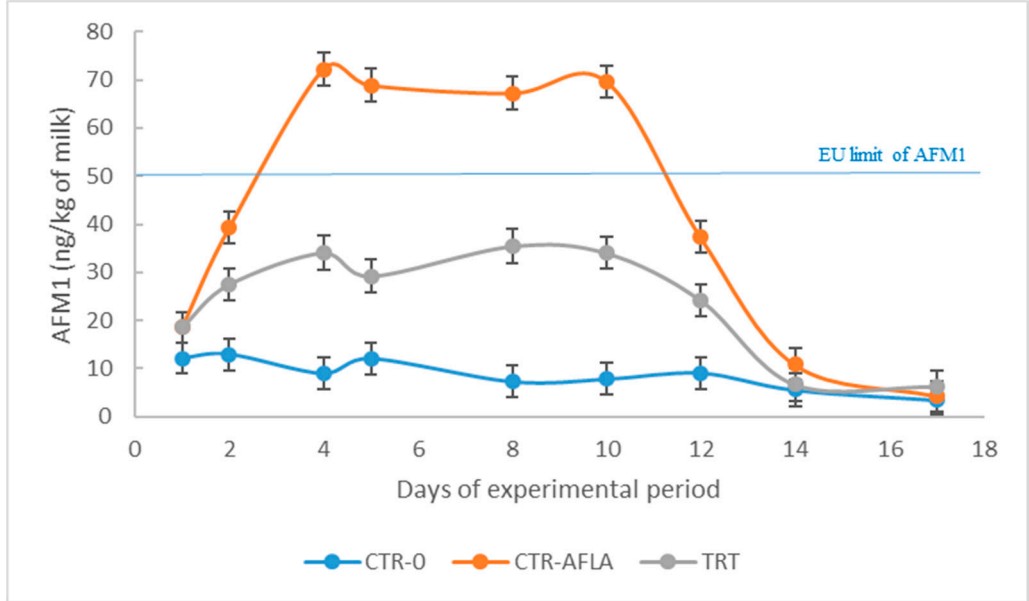

**Figure 1.** AFM1 concentration in the milk of Holstein cows in that received CTR-0, CTR-AFLA, and the TRT diets during each day of experimental period. The "plateau" (steady state) period was from day 4 to day 10.

**Table 3.** Least squares means and associated standard errors of the mean (pooled SEM) of AFB1 intake, milk AFM1, and carry-over of AFB1 into milk of Holstein cows that received CTR-0, CTR-AFLA, and TRT diets during each experimental period. Values with different superscript letters in the same group and row were significantly different ($p < 0.05$).

| Items | Units | Treatment | | | Period | | | Pooled SEM | *p* of the Model | | | | |
|---|---|---|---|---|---|---|---|---|---|---|---|---|---|
| | | CTR-0 | CTR-AFLA | TRT | 1 | 2 | 3 | | Latin Square | Period | Treatment (T) | Day (D) | D * T |
| AFB1 Intake * | µg/cow/d | 11.204 [c] | 42.461 [a] | 39.197 [b] | 31.301 [a,b] | 34.039 [a] | 27.522 [b] | 16.0351 | 0.333 | <0.05 | <0.05 | 0.457 | 0.911 |
| Milk, AFM1 * | ng/kg milk | 8.81 [c] | 43.16 [a] | 24.01 [b] | 32.40 [a] | 22.51 [b] | 21.08 [b] | 10.065 | 0.597 | <0.05 | <0.05 | <0.05 | <0.05 |
| Milk, AFM1 ** | ng/kg milk | 8.90 [c] | 69.51 [a] | 33.96 [b] | 47.22 [a] | 36.44 [b] | 28.71 [c] | 27.958 | 0.688 | <0.05 | <0.05 | 0.876 | 0.454 |
| AFM1 excretion * | µg/cow/d | 0.254 [c] | 1.230 [a] | 0.683 [b] | 0.974 [a] | 0.643 [b] | 0.549 [b] | 0.01076 | 0.393 | <0.05 | <0.05 | <0.05 | <0.05 |
| AFM1 excretion ** | µg/cow/d | 0.262 [c] | 1.975 [a] | 0.969 [b] | 1.440 [a] | 1.038 [b] | 0.729 [c] | 0.0292 | 0.536 | <0.05 | <0.05 | 0.499 | 0.206 |
| Carry Over ** | % | 2.23 [b] | 4.73 [a] | 2.51 [b] | 4.16 [a] | 3.07 [b] | 2.25 [c] | 0.323 | 0.562 | <0.05 | <0.05 | 0.666 | 0.409 |

* For the entire intoxication period (day 1 to day 10 of the experimental period; Figure 1). ** For the plateau (steady-state) period (day 4 to day 10 of the experimental period; Figure 1).

*3.4. Body Weight, Body Condition Score, and Blood Parameters*

Table 4 shows the relationships of diet with BW, BCS, and blood parameters. The BCS and BW did not differ among the treatments and the overall averages were 2.23 points (BCS) and 713.8 kg (BW). However, the BW had a tendency ($p = 0.091$) to be lower in the TRT group than in the other groups. Furthermore, some plasma biomarkers had tendencies to be different ($p < 0.10$) in TRT group (total cholesterol, Mg, paraoxonase, and total bilirubin); all of these parameters were greater in the TRT group than in the CTR-0 group, suggesting a possible benefit of the TRT diet. However, these differences were all small and unrelated to metabolic dysfunctions, except for total bilirubin. The presence of a higher level of total bilirubin in the TRT group suggests a possible interference in the release of liver enzymes responsible for bilirubin clearance. Consequently, the smectite clay binder had a beneficial effect, even though it modified some plasma parameters.

**Table 4.** Least squares means and associated standard errors of the mean (pooled SEM) of BW, BCS, and blood parameters of Holstein cows that received CTR-0, CTR-AFLA, and TRT diets during each experimental period. Values with different superscript letters in the same group and row were significantly different ($p < 0.05$).

| Items | Units of Measurements | Treatment | | | Period | | | Pooled SEM | $p$ of the Model | | |
|---|---|---|---|---|---|---|---|---|---|---|---|
| | | CTR-0 | CTR-AFLA | TRT | 1 | 2 | 3 | | Latin Square | Period | Treatment |
| BCS | Scale 1–4 | 2.30 | 2.29 | 2.28 | 2.33 [a] | 2.34 [a] | 2.21 [b] | 0.003 | 0.397 | <0.05 | 0.910 |
| BW | kg | 720.2 | 718.6 | 702.6 | 715.0 | 715.8 | 710.6 | 117.20 | 0.071 | 0.7972 | 0.097 |
| PCV | L/L | 0.332 | 0.352 | 0.348 | 0.336 | 0.343 | 0.353 | 0.0002 | 0.132 | 0.3044 | 0.201 |
| Glucose | mmol/L | 4.50 | 4.57 | 4.55 | 4.53 | 4.51 | 4.58 | 0.026 | 0.324 | 0.8683 | 0.857 |
| Total Cholesterol | mmol/L | 5.24 | 5.49 | 5.56 | 5.75 [a] | 5.35 [b] | 5.18 [b] | 0.031 | 0.841 | <0.05 | 0.079 |
| Urea | mmol/L | 5.89 | 5.93 | 5.79 | 5.55 [b] | 6.61 [a] | 5.46 [b] | 0.199 | 0.327 | <0.05 | 0.916 |
| Calcium | mmol/L | 2.50 | 2.52 | 2.52 | 2.50 | 2.55 | 2.48 | 0.003 | 0.342 | 0.2614 | 0.901 |
| Magnesium | mmol/L | 1.07 | 1.16 | 1.13 | 1.17 [a] | 1.11 [a,b] | 1.08 [b] | 0.002 | 0.196 | <0.05 | 0.081 |
| Zinc | µmol/L | 13.66 | 15.00 | 14.57 | 15.75 [a] | 14.40 [a,b] | 13.08 [b] | 1.949 | 0.934 | 0.0768 | 0.462 |
| Ceruloplasmin | µmol/L | 2.47 | 2.67 | 2.64 | 2.50 | 2.59 | 2.69 | 0.033 | 0.915 | 0.4450 | 0.321 |
| Total Protein | g/L | 83.3 | 84.6 | 84.0 | 85.0 | 84.0 | 82.9 | 1.85 | <0.05 | 0.1604 | 0.507 |
| Albumin | g/L | 39.3 | 39.7 | 39.5 | 40.3 [a] | 39.4 [b] | 38.9 [b] | 0.24 | 0.926 | <0.05 | 0.488 |
| Globulin | g/L | 44.0 | 44.8 | 44.5 | 44.8 | 44.6 | 44.0 | 1.29 | <0.05 | 0.6399 | 0.666 |
| AST/GOT | U/L | 84.8 | 85.5 | 84.6 | 86.9 | 80.6 | 87.4 | 23.36 | 0.698 | 0.1602 | 0.965 |
| GGT | U/L | 28.1 | 28.9 | 29.2 | 29.8 [a] | 28.4 [b] | 28.1 [b] | 0.76 | 0.444 | <0.05 | 0.290 |
| Total Bilirubin | µmol/L | 1.56 | 1.61 | 2.04 | 1.72 [a,b] | 1.45 [b] | 2.05 [a] | 0.074 | 0.371 | <0.05 | 0.069 |
| Haptoglobin | g/L | 0.363 | 0.414 | 0.403 | 0.317 | 0.429 | 0.436 | 0.0176 | 0.748 | 0.4438 | 0.871 |
| Paraoxonase | U/L | 103.5 | 108.2 | 105.0 | 106.4 | 104.6 | 105.7 | 6.98 | 0.196 | 0.6931 | 0.098 |
| ROMt | (mg $H_2O_2$/dL) | 13.13 | 14.33 | 13.67 | 13.29 | 13.87 | 13.97 | 1.003 | 0.892 | 0.6419 | 0.324 |
| Myeloperoxidase | U/L | 268.5 | 285.9 | 279.8 | 286.5 | 257.7 | 290.0 | 711.47 | 0.718 | 0.2573 | 0.695 |
| FRAP | µmol/L | 160.1 | 157.8 | 152.4 | 148.7 | 157.5 | 219.7 | 119.75 | 0.668 | 0.215 | 0.654 |

Abbreviations: BCS, Body Condition Score; BW, body weight; PCV, plasma cell volume; AST/GOT, glutamic oxaloacetic transaminase; GGT, gamma glutamyl transferase; ROMt, total reactive oxygen metabolites; FRAP, ferric ion reducing antioxidant power.

## 4. Discussion

The aim of this study of multiparous mid-lactation Holstein cows was to assess the effects of dietary supplementation with a commercially available smectite clay to an AFB1 challenge on the excretion of AFM1 in dairy milk, and on the performance parameters, blood chemistry, and liver inflammatory markers of the cows. We hypothesized that the smectite clay supplement would lead to a reduced excretion of AFM1 into milk, and the maintenance of well-being, health-status, and performance parameters. Our results are consistent with this hypothesis.

Previous studies have described the effects of different clay feed additives on AF excretion and AF transfer from feed to milk [49,50]. In our experimental conditions, the addition of a smectite clay (TOXO-MX) into a traditional lactation diet given to multiparous lactating Holstein cows reduced the milk AFM1 concentration by 64.8% and had a carry-over reduction of 47.0%. Previous studies reported that cows in early lactation can excrete 3.8–6.2% of dietary AFB1 as AFM1 (the main metabolite) into their milk, less than cows in late lactation (1.8–2.5%) [51]. However, the rate of biotransformation depends on animal status and nutritional and physiological factors, such as type of diet, rate of ingestion, digestion rate, animal health, biotransformation capacity of the liver, and dairy animal production [5,52]. Our results (Table 3) indicated that the values in milk were 4.73% for the CTR-AFLA group and 2.51% for the TRT group during the "plateau" (steady state) period, from day 4 to day 10 of the experimental period.

Sulzberger et al. [28] previously evaluated the impact of different concentrations of dietary clay (0.5%, 1%, and 2% of dietary DMI) after an AF challenge (100 μg AFB1/kg DMI) on several dairy milk parameters. They reported that the use of the clay into the diet reduced the carry-over by an average of 33.6%. Xiong et al. [53] evaluated the effects of a dietary adsorbent (a hydrated calcium sodium aluminosilicate at a level of 60 g/cow/day) on milk AFM1 content and the health of lactating dairy cows exposed to long-term AFB1 challenge (20 μg AFB1/kg of diet dry matter). They reported that the different treatments had no significant differences in: DMI; milk yield; percentages of milk protein, milk fat and lactose; somatic cell counts. However, the adsorbent significantly reduced the milk AFM1 concentration (0.19 vs. 0.13 μg/kg) and transfer rate (1.38 vs. 0.89%), similar to our findings. Another study examined cows fed a TMR diet with 55 μg/kg AFB1 and reported that multiple products reduced the dairy milk concentration of AFM1 (activated carbon: 5.4%, esterified glucomannan: 59%, calcium bentonite: 31%, and three hydrated sodium calcium aluminosilicate [HSCAS] products: 65%, 50%, and 61%) [54]. The addition of bentonite (AB-20) to the diet of cattle reduced the AFM1 concentration by 60.4% in the milk of cows fed AF of 80 μg/kg [55]. Maki et al. [56] examined the effect of a clay feed additive at 0.5 and 1% of dietary DM in response to an AF challenge (daily doses of 100 μg/kg estimated DMI via a top-dressed supplement in rice powder containing 758 mg of AFB1/kg of weight). Their measurements of milk AFM1 concentration indicated that 0.5% of clay feed reduced the level by 51.3% and that 1% clay feed reduced the level by 69.7%. Interestingly, our findings also showed that the CTR-0 and TRT diets led to comparable carry-over values, even though there were clear differences in the absolute concentrations (Table 3). This topic deserves further investigation.

Some previous studies reported no significant changes in DMI and milk yields of cows following an AF challenge with the addition of clay-based feeding systems. However, we found lower ($p < 0.05$) DMI values for the CTR-AFLA and TRT groups than the CTR group (Table 2). There were also small but significant changes ($p < 0.05$) in 3.5% FCM/DMI, ECM/DMI, and milk/DMI (Table 2). The reduced DMI in the CTR-AFLA and TRT groups is in agreement with the modest increase ($p > 0.05$) in total bilirubin in these two groups. Although not significant, the increased bilirubin level suggests impaired hepatic function [57], despite our usage of a low dosage of AFs seem to suppose a co-contamination of diets with other mycotoxins. Overall, our failure to detect other differences in blood biochemical indicators may be attributable to the low amount of dietary AFB1 (~20 μg/kg), as suggested previously [58,59]. For example, there is evidence that mycotoxins disrupt some functions of the immune system, and dairy cows fed AFB1 display increased innate immune responses [60]. In this regard, the haptoglobin level (an indicator of innate immune stress) increased after endotoxin challenge. However, our groups had

no significant differences in haptoglobin concentrations (Table 4), likely because our AFB1 dose was too low to affect this parameter. Therefore, more measurements and studies of immune function (such as the number and activity of lymphocytes or release of specific cytokines) are needed to provide a better understanding of the findings presented here. Interestingly, our results were similar for the CTR-AFLA and TRT groups when considering the modest increase in total bilirubin and the significantly lower DMI values.

Overall, our data suggest that AFs can impair the health statuses of animals before absorption. In fact, as extensively reviewed by Grenier and Applegate [61], intestinal cells are first exposed to mycotoxins, often at higher concentrations than the cells of other tissues. These fungal metabolites potentiate intestinal inflammation, although there is little known about their effect on intestinal microbiota [61]. The maintenance of a healthy gastro-intestinal tract ensures that nutrients are adequately absorbed, provides protection against pathogens through the immune system, and maintains an adequate balance of endogenous microflora [62]. Therefore, in our experimental conditions, AFs may have compromised the integrity of the intestinal mucosa, thus leading to reduced nutrient absorption. Ogunade et al. [63] studied the effects of adding three mycotoxin-sequestering agents to diets contaminated with AFB1 (75 µg/kg of dietary DMI) on the milk AFM1 level and immune status of dairy cows. Their fluorescence assay indicated greater levels of two leukocyte adhesion molecules (L-selectin and β-integrin) on neutrophils of cows that received diets containing yeast cell culture and sodium bentonite. This is of great interest, because blood neutrophils are the first line of defense in the innate immune system, and elevated levels suggest migration to intestinal cells that were exposed to the toxin. Our results (Table 4) indicated no significant differences in the BCS of the different groups, possibly because the short duration of the experimental period and the use of dairy cows with mid-lactation status [28].

Our analysis of the amount of AFM1 in dairy milk following the three different treatments indicated that the CTR diet led to an AFM1 level of 0.0089 µg/kg during the steady state period, much lower than the 0.05 µg/kg threshold established by the EU. In contrast, the CTR-AFLA group had an unacceptable level of AFM1 contamination (0.069 µg/kg), and the addition of the smectite clay led to an AFM1 contamination of 0.034 µg/kg (below the EU limit). Therefore, the commercial smectite clay used in this study reduced the risk of AF contamination. We also examined the effect of diet on milk coagulation properties (MCPs). This topic has received much attention in academic dairy science and industry, because the amount of milk used to manufacture cheese is increasing worldwide, and there is evidence of a correlation between MCPs and cheese characteristics (i.e., processing, yield, and quality) [64].

The main sources of variation in MCP are classified as genetic (species, breed, major genes, and polygenes of dairy cows) and environmental (climate, season, farming system, feeding, hygiene, health, milking conditions) [62]. The most commonly measured parameters related to MCP are the lactodynamographic parameters: (a) rennet coagulation time (RCT or r, min); (b) time to curd firmness of 20 mm (k20, min); (c) curd firmness 30 min after enzyme addition (a30, mm), defined by the width of the graph when the test usually ends. The a30 value is often highly dependent on the r value, and is affected by environmental, physiological and genetic factors [62]. Thus, a longer time to coagulation indicates less time available for curd firming and lower final firmness. Importantly, our findings suggested that the presence of dietary TOXO-MX did not significantly affect the milk quality parameters and coagulation properties. In our experimental conditions, the r was 18.41 min and the a30 was 29.51 mm. The combinations of the three MCP parameters can be used to classify milk as optimal (Type A), suboptimal (Types B, C, D), defective (Types E, F, DD), and non-coagulating (Type FF). Therefore, the MCP parameters in our experimental conditions were Type B, with an RCT/a30 coefficient of 0.62.

## 5. Conclusions

The addition of AFB1 into experimental diets reduced the DMI of dairy cows. However, the MY (corrected or uncorrected) was not affected by the different diets, so the highest feed efficiency was in

the TRT group. The levels of MY and DMI reported here could have been affected by the specific animal housing conditions, in particular the 4-week adaptation period. The AFM1 level in milk reached a "plateau" from day 4 to day 10 of the intoxication period. During this time, the addition of TOXO-MX reduced milk AFM1 concentration by 64.8% and reduced carry-over by 47.0%. Overall, the CTR-0 and TRT diets had very comparable carry over values, even though the absolute concentrations were very different. The plasma biomarkers we measured were all within the reference intervals for dairy cows in mid-lactation. Thus, the influence at the inflammo-metabolic level appeared to be attenuated or almost absent. It is likely that a prolonged administration of this mycotoxin could lead to more severe consequences at the inflammo-metabolic level. We hypothesize that severe consequences may occur at a higher dosage than used in our experiments and/or in the presence of other mycotoxins that affect the integrity of the gastro-intestinal epithelium.

**Author Contributions:** Conceptualization, A.G., S.v.K., and E.T.; Methodology, A.G., F.P.C., S.P., A.M., and E.T.; Formal analysis, A.G., F.P.C., S.v.K., Y.H., and E.T.; Investigation, F.P.C. and S.P.; Data curation, A.G., F.P.C., and E.T.; Writing—original draft preparation, A.G. and G.R.; Writing—review and editing, A.G., F.P.C., S.v.K., Y.H., and E.T.; Project administration, A.G. and E.T.; Funding acquisition, A.G. All authors have read and agreed to the published version of the manuscript.

**Funding:** This research was funded by Nutreco R & D, Amersfoort, The Netherlands and by the Fondazione Romeo ed Enrica Invernizzi (Milan, Italy).

**Acknowledgments:** The authors are grateful to Silvia Colombi for their assistance with animal housing and Annalisa Ferrari and Gloria Luzzani for their assistance with laboratory analyses.

**Conflicts of Interest:** The authors Sandra van Kuijk and Yanming Han are employees of Nutreco R & D; they played no role in the collection, analysis or interpretation of data. All other authors declare no potential conflicts of interest.

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
