# Peer review of "Effect of a Commercial Bentonite Clay (Smectite Clay) on Dairy Cows Fed Aflatoxin-Contaminated Feed"

_2624-862X, doi:10.3390/dairy1020009_

Round 1

Reviewer 1 Report

Dear Authors,

I have revised the manuscript entitled “Effect of a commercial bentonite (smectite clay) clay on dairy cows fed mycotoxin contaminated feed”.

General comments

This study is very interesting and worth of publication. However, some minor questions need to be answered. For example, the tested aflatoxins levels are not informed or supported with literature in the Introduction. See specific comments.

Specific comments

Abstract

Line 21: Inform the tested Aflatoxin conentrations

Introduction

In the complete Introduction section, when citing studies effects of aflatoxins (acute and chronic), inform the levels in diets.

What are the real critical levels for carry-over, knowing that high yielding cows have a higher carry-over than those producing less milk?

Line 44: Please, give the original reference.

Materials and Methods

Line 119 “…AFB1 contamination levels of 1.76±2.06 or 17.53±6.55, respectively.” ppb?

Discussion

When citing studies on aflatoxins, inform the tested levels for a correct comparison of data.

Author Response

REVIEW 1 REPORT

Dear Authors,

 I have revised the manuscript entitled “Effect of a commercial bentonite (smectite clay) clay on dairy cows fed mycotoxin contaminated feed”. 

General comments

This study is very interesting and worth of publication. However, some minor questions need to be answered. For example, the tested aflatoxins levels are not informed or supported with literature in the Introduction. See specific comments.

Authors: We would like to thank the reviewer for having appreciated this work. The Introduction section has been modified accordingly, by adding more supporting sentences. Thank you for your help in improving the overall quality of this manuscript.

Specific comments

Abstract

Line 21: Inform the tested Aflatoxin concentrations.

Authors: The Abstract has been revised, accordingly: "In particular, 3 cows each received the CTR-0 diet (total mixed ration [TMR] with normal corn meals), the CTR-AFLA diet (CTR-0 diet with 17.53±6.55 µg/kg DM AFBI), or the TRT diet (CTR-AFLA diet with 100±1 g/cow/day of smectite clay). The AFB1 level was 0.63±0.50 µg/kg DM in the CTR-0 diet, 2.28±1.42 µg/kg DM in the CTR-AFLA diet, and 2.13±1.11 µg/kg DM in the TRT diet."

Introduction

In the complete Introduction section, when citing studies effects of aflatoxins (acute and chronic), inform the levels in diets. What are the real critical levels for carry-over, knowing that high yielding cows have a higher carry-over than those producing less milk?

Authors: The info required have been added to the Introduction section. Please, check lines 65-79 of the revised Manuscript.

Line 44: Please, give the original reference.

Authors: Indeed, prior to 1985, the Food and Agriculture Organization (FAO) estimated global food crop contamination with mycotoxins to be 25%. However, as strongly discussed by Eskola et al. (2019) (https://doi.org/10.1080/10408398.2019.1658570) the origin of this statement is largely unknown and not fully detailed in the original reference (i.e. Park et al. 1999). Accordingly, Eskola et al. (2019) reviewed the relevant literature about this topic in order to assess the validity of this statement. Data of around 500,000 analyses from the European Food Safety Authority and large global survey for aflatoxins, fumonisins, deoxynivalenol, T-2 and HT-2 toxins, zearalenone and ochratoxin A in cereals and nuts were examined. Using different limit of detection, the lower and upper regulatory limits of European Union (EU) legislation and Codex Alimentarius standards, the mycotoxin occurrence was estimated. Overall, the current mycotoxin occurrence above the EU and Codex limits appeared to confirm the FAO 25% estimation, while this figure greatly underestimated the occurrence above the detectable levels (up to 60–80%). The high occurrence was likely explained by a combination of the improved sensitivity of analytical methods and impact of climate change. It is of immense importance that the detectable levels are not overlooked as through diets, humans are exposed to mycotoxin mixtures which can induce combined adverse health effects.

Materials and Methods

Line 119 “…AFB1 contamination levels of 1.76±2.06 or 17.53±6.55, respectively.” ppb?

Authors: Yes, ppb. The sentence has been revised, accordingly.

Discussion

When citing studies on aflatoxins, inform the tested levels for a correct comparison of data.

Authors: done. Thank you for pointing it out.

Reviewer 2 Report

Dear editor,

Thank you very much for sending the manuscript for reviewing. The current work describes the effect of well-known commercial clay to bind aflatoxins as a tool for mycotoxins reduction or decontamination. Furthermore, the authors have evaluated the performance of this commercial product. The manuscript is well written, however it needs to be doubled checked again. The experimental design and statistical analysis seem sound. I have some questions and suggestions which are indicated below.

When was this experiment done?

Did the author check the natural (co-)occurrence of other mycotoxins in the animal feeds?

Title

You should change “dairy cow fed mycotoxin contaminated feed” into “dairy cow fed aflatoxin contaminated feed”.

Abstract

Line 16, “aflatoxin binders” instead of “aflatoxin challenge”.

Line 17, the carry-over of aflatoxin B1 through detecting aflatoxin M1 (main metabolite) in milk.

Line 22, what is the dose of AFB1?

Since you added abbreviation in the abstract then follow the same for Aflatoxin B1 and M1 in the abstract. Also add the word “dairy” before the word “milk” in your manuscript.

Suggested to add the following reference

Adballah et al., 2015. Occurrence, prevention and limitation of mycotoxins in feeds.

Line 43, this is an old info otherwise add the update reference for that.

Line 48, replace the reference 3 with the following

Streit, et al., 2013. Multi-mycotoxin screening reveals the occurrence of 139 different secondary metabolites in feed and feed ingredients. 

And Abdallah et al., 2017 Occurrence of multiple mycotoxins and other fungal metabolites in animal feed and maize samples from Egypt using LC‐MS/MS.

Line 50 this is not true for all mycotoxins! It is correct for “most” of the known mycotoxins.

Line 62, replace extremely toxic with hepatocarcinogenic and refer to IARC.

Line 63, suggest adding Abdallah et al, 2019 Mycotoxin detection in maize, commercial feed, and raw dairy milk samples from Assiut City, Egypt.

Line 65, delete “The most used techniques” otherwise add for example a figure showing the percentage this approach in comparison to other pre-harvest approach and in which crop and for which toxigenic fungal species.

harvested feed? Sounds a strange term, replace with another convenient word.

Line 71, Replace ppb with µk/kg and apply this to the whole manuscript.

The last part need to be rearranged and to consistent. There is kind of a jumping from one stiry to another. For example in line 62 you mentioned that AFs are extremely toxic then the mitigation then regulation then back to the clinical signs which should be above. The introduction part seems good but need rearrangement.

Line 83, you went again to the mitigation strategy which u started to talk about in 66 and from line 74 until line 82 you described the toxic effect of AFs. So it is kind of jumping in my opinion.

Material and Methods

Line 151, replace aflatoxins with AF since you have already inserted above. Make sure that this is done throughout the manuscript for all the abbreviation.

Line (41:59, vol/vol)

Line 157, what kind of vial? Glass vial, plastic vial or HPLC vial?

Line 160, what is BCS ??

Line 165, Their health status was monitored daily, how? Plz indicate that.

Some health problems, like what ?

174 at each milking time.

Line 174, Representative milk samples, one litre two three ???

Line 184, what do you mean by “analyzed as described previously” ?? in 185, you described and you added different reference though. Check this part.

Line (25:75, vol/vol).

Line 194, what do you mean by AFMs?? Is not mentioned in your manuscript. Also line 197.

Line 196, using “an” isocratic elution

Line 196, using isocratic elution with water and a mixture of acetonitrile: methanol (17: 29, v/v) with a 59: 41 (v/v) ratio, at flow rate of 1 ml/min. change to “using an isocratic elution of 59: 41 (v/v) of mobile phase A (water) and mobile B (a mixture of acetonitrile: methanol (17: 29, v/v)), respectively at a flow rate of 1 ml/min”.

Line 198, UV lamp at 254 nm, plz add the brand name, company and the country. Check all the material and method section for that as well.

Line 203, what is AFBs? You are talking about AFB1 right? Then fix it.

Line 209, a small amount of blood , how many ml ??

Line 211, the remaining blood, what is the volume of the reminding volume you are claiming ?

Results

Line 259, Did the author check the natural (co-)occurrence of other mycotoxins in the animal feeds?

No other aflatoxins were detected in both TMRs. How? Using the same analytical method? And did you check all the other feed samples?

Line 325, Anyway, these plasma modifications are light and probably without clinical consequences for dairy cows. Rewrite, this is not scientific English

Under table 2 and other tables, add the full name of each abbreviation included in the table.

Discussion

Line 333, why did you write aflatoxin again in full term? Double check the abbreviation throughout the whole manuscript.

339, add references >> the literature if full of that

Suggest adding Abdallah et al, 2019 Mycotoxin detection in maize, commercial feed, and raw dairy milk samples from Assiut City, Egypt.

Line 344 and 345, this is already mentioned in the introduction; please avoid the repetition of the same info.

The discussion part is quite long and therefore the reader will get lost easily.

Suggest shorting the discussion and delete the unnecessary info especially the information that is known to the reader in the field of mycotoxins.

Author Response

REVIEW 2 REPORT

Dear editor,

Thank you very much for sending the manuscript for reviewing. The current work describes the effect of well-known commercial clay to bind aflatoxins as a tool for mycotoxins reduction or decontamination. Furthermore, the authors have evaluated the performance of this commercial product. The manuscript is well written, however it needs to be doubled checked again. The experimental design and statistical analysis seem sound. I have some questions and suggestions which are indicated below.

Authors: We would like to thank the reviewer for having appreciated this work.

When was this experiment done?

Authors: Done. Please see line 114.

Did the author check the natural (co-)occurrence of other mycotoxins in the animal feeds? 

Authors: The reviewer asked to check for co-precense of other mycotoxins and we strongly agree for this. We analyzed corn meals used in this experiment only for aflatoxins contamination and this is mainly due the fact the contaminated corn meals used in current experiment was obtained by in field crop inoculation with a mycotoxigenic A. flavus strain provided by pathologists of Dipartimento di Scienze delle produzioni vegetali sostenibili (Di.Pro.Ve.S.) of Università Cattolica del Sacro Cuore. Consequently, a contamination by Fusarium-, Alternaria-, Penicillum- or other fungal strains-produced mycotoxin was unexpected. We added this information in the main text. Please, see lines 133-135. At this moment, we have two problems in doing the analysis. Our labs are closed for summer vacation and the time for reviewing the manuscript are very strictly. So, we decided to send the revised version of the manuscript without re-analyze corn meal for other mycotoxins. 

Title

You should change “dairy cow fed mycotoxin contaminated feed” into “dairy cow fed aflatoxin contaminated feed”.

Authors: Thank you for the suggestion. The title has been modified, accordingly.

Abstract

Line 16, “aflatoxin binders” instead of “aflatoxin challenge”.

Authors: done.

Line 17, the carry-over of aflatoxin B1 through detecting aflatoxin M1 (main metabolite) in milk.

Authors: Revised, accordingly.

Line 22, what is the dose of AFB1?

Authors: The Abstract has been revised, accordingly: " In particular, 3 cows each received the CTR-0 diet (total mixed ration [TMR] with normal corn meals), the CTR-AFLA diet (CTR-0 diet with 17.53±6.55 µg/kg DM AFBI), or the TRT diet (CTR-AFLA diet with 100±1 g/cow/day of smectite clay). The AFB1 level was 0.63±0.50 µg/kg DM in the CTR-0 diet, 2.28±1.42 µg/kg DM in the CTR-AFLA diet, and 2.13±1.11 µg/kg DM in the TRT diet.”

Since you added abbreviation in the abstract then follow the same for Aflatoxin B1 and M1 in the abstract. Also add the word “dairy” before the word “milk” in your manuscript.

Authors: done. Thank you for pointing it out.

Suggested to add the following reference

Adballah et al., 2015. Occurrence, prevention and limitation of mycotoxins in feeds.

Line 43, this is an old info otherwise add the update reference for that.

Authors: we have added the reference Adballah et al. 2015.

Line 48, replace the reference 3 with the following

Streit, et al., 2013. Multi-mycotoxin screening reveals the occurrence of 139 different secondary metabolites in feed and feed ingredients. 

And Abdallah et al., 2017 Occurrence of multiple mycotoxins and other fungal metabolites in animal feed and maize samples from Egypt using LC‐MS/MS.

Authors: done.

Line 50 this is not true for all mycotoxins! It is correct for “most” of the known mycotoxins.

Authors: revised, accordingly.

Line 62, replace extremely toxic with hepatocarcinogenic and refer to IARC.

Authors: done.

Line 63, suggest adding Abdallah et al, 2019 Mycotoxin detection in maize, commercial feed, and raw dairy milk samples from Assiut City, Egypt.

Authors: done.

Line 65, delete “The most used techniques” otherwise add for example a figure showing the percentage this approach in comparison to other pre-harvest approach and in which crop and for which toxigenic fungal species.

Authors: We would like to thank the reviewer for the suggestion. However, considering that we have already 5 among tables and figures, we preferred to not add another figure, in order to facilitate the readability of the manuscript. Indeed, we have modified the misleading sentence, accordingly.

harvested feed? Sounds a strange term, replace with another convenient word.

Authors: done.

Line 71, Replace ppb with µk/kg and apply this to the whole manuscript.

Authors: done.

The last part need to be rearranged and to consistent. There is kind of a jumping from one stiry to another. For example in line 62 you mentioned that AFs are extremely toxic then the mitigation then regulation then back to the clinical signs which should be above. The introduction part seems good but need rearrangement.

Line 83, you went again to the mitigation strategy which u started to talk about in 66 and from line 74 until line 82 you described the toxic effect of AFs. So it is kind of jumping in my opinion.

Authors: The Introduction section has been deeply modified, mainly when considering the last part. Thank you for pointing it out.

Material and Methods

Line 151, replace aflatoxins with AF since you have already inserted above. Make sure that this is done throughout the manuscript for all the abbreviation.

Authors: done.

Line (41:59, vol/vol)

Authors: done.

Line 157, what kind of vial? Glass vial, plastic vial or HPLC vial?

Authors: We used HPLC vials. The info has been added to the text.

Line 160, what is BCS??

Authors: we have added the meaning (i.e., body condition score).

Line 165, Their health status was monitored daily, how? Plz indicate that.

Authors: The health status was monitored daily by farm’s vet. We added this information into revised version of the manuscript. Please, see lines 186-187.

Some health problems, like what?

Authors: We added this information on line 191-192.

174 at each milking time.

Authors: revised, accordingly.

Line 174, Representative milk samples, one litre two three ???

Authors: We sampled 100 ml of milk assuring to be representative of milk produced at each milking. We added this information in the revised version. Please, see line 196.

Line 184, what do you mean by “analyzed as described previously” ?? in 185, you described and you added different reference though. Check this part.

Authors: We have revised the misleading sentence. Thank you for pointing it out.

Line (25:75, vol/vol).

Authors: done.

Line 194, what do you mean by AFMs?? Is not mentioned in your manuscript. Also line 197.

Authors: We apologize for the misleading terms. Overall, there was an error in the disposition of the M&M sections regarding analysis of aflatoxins in feed and milk samples. We have revised the paragraphs 2.2 and 2.5, accordingly.

Line 196, using “an” isocratic elution

Authors: revised, accordingly.

Line 196, using isocratic elution with water and a mixture of acetonitrile: methanol (17: 29, v/v) with a 59: 41 (v/v) ratio, at flow rate of 1 ml/min. change to “using an isocratic elution of 59: 41 (v/v) of mobile phase A (water) and mobile B (a mixture of acetonitrile: methanol (17: 29, v/v)), respectively at a flow rate of 1 ml/min”.

Authors: revised, accordingly.

Line 198, UV lamp at 254 nm, plz add the brand name, company and the country. Check all the material and method section for that as well.

Authors: we added this information on lines 177-178.

Line 203, what is AFBs? You are talking about AFB1 right? Then fix it.

Authors: We would thank this review for caching this mistake. We referred to aflatoxins, abbreviated as AFs and not AFBs. Please, see line 176.

Line 209, a small amount of blood , how many ml ??

Authors: we added this information in specific sentence. Please, see line 227.

Line 211, the remaining blood, what is the volume of the reminding volume you are claiming?

Authors: we added this information in specific sentence. Please, see line 229.

 Results

Line 259, Did the author check the natural (co-)occurrence of other mycotoxins in the animal feeds?

Authors: please, refer to previous comment.

No other aflatoxins were detected in both TMRs. How? Using the same analytical method? And did you check all the other feed samples?

Authors: We analyzed aflatoxins contamination with the same method in the corn meals and TMR. We added this information for readers on lines 165-166.

Line 325, Anyway, these plasma modifications are light and probably without clinical consequences for dairy cows. Rewrite, this is not scientific English

Authors: We have removed the misleading sentence.

Under table 2 and other tables, add the full name of each abbreviation included in the table.

Authors: done.

Discussion

Line 333, why did you write aflatoxin again in full term? Double check the abbreviation throughout the whole manuscript.

Authors: we have revised the abbreviations throughout the manuscript.

339, add references >> the literature if full of that. Suggest adding Abdallah et al, 2019 Mycotoxin detection in maize, commercial feed, and raw dairy milk samples from Assiut City, Egypt.

Authors: we have removed this part of the Discussion section, in order to ameliorate the Manuscript readability.

Line 344 and 345, this is already mentioned in the introduction; please avoid the repetition of the same info.

Authors: the sentence has been removed, accordingly.

The discussion part is quite long and therefore the reader will get lost easily.

Suggest shorting the discussion and delete the unnecessary info especially the information that is known to the reader in the field of mycotoxins.

Authors: The Discussion section has been shortened and revised, accordingly.

Round 2

Reviewer 2 Report

The manuscript is significantly improved and the authors could respond to all the comments. 

I have no further comments. Good luck with your future research!